# GIAHS as an Instrument to Articulate the Landscape and Territorialized Agrifood Systems—The Example of La Axarquía (Malaga Province, Spain)

Rocío Silva-Pérez and Gema González-Romero *

Department of Human Geography, University of Sevilla, 41004 Sevilla, Spain; rsilva@us.es
* Correspondence: gemagonzalez@us.es

**Abstract:** The theoretical literature makes a connection between the notions of landscape and territorialised agrifood systems, but these connections are rarely specified or explained. Their consideration in development proposals requires the relationship between the two and their magnitude to be made explicit. This article enquires into this and explores its programmatic forecasts from both the theoretical and empirical perspectives. An epistemological and regulatory review points to the FAO Globally Important Agriculture Heritage System (GIAHS) programme as the instrument that articulates the logic of landscape and territorialised agrifood in marginal rural areas. The empirical part of the study focuses on the dried grape (raisin) agrifood system in Axarquía (Malaga province, Spain) GIAHS since 2018. Territorial recognition and semi-structured interviews have enabled a deep study of the praxis of these connections. Axarquía is seen to be an excellent laboratory of the synergies between territorialised agrifood system synergies and landscapes. GIAHS is substantiated as a virtual instrument that can contribute to agriculture-based territorial development. It acts as a stimulus to development and combines forces through territorial governance processes. It highlights the value of agriculture as a cornerstone of development, examines agricultural functionalities in detail, and gives meaning to agricultural landscapes.

**Keywords:** territorialised agrifood systems; landscapes; Globally Important Agriculture Heritage System; territorial governance; Axarquía

## 1. Introduction

The notions of 'territorialised agrifood systems' and 'landscapes' enjoy great prestige. The theoretical literature connects the two, but these links are rarely made explicit or sufficiently explained according to their different epistemological and regulatory foundations. In disciplinary terms, they are increasingly being addressed in multiple ways that do not always coincide with their understanding and analysis; the same is true of their institutional and regulatory frameworks. In addition, in recent years, landscape and territorialised agrifood system have resorted to articulating proposals for local territorial development underpinned by citizen involvement and governance [1,2].

Agriculture is recognised not only to supply food and raw products but also to provide society with public goods (re: the environment, landscape, heritage, etc.) [3–6]. In some cases, these are agrarian systems of high natural value [7,8] considered essential for maintaining biodiversity and mitigating climate change [9] and, in others cases, with exceptional landscapes listed as UNESCO World Heritage sites [10]. Both of these examples refer to agriculture of great heritage value but a small market price in areas blighted by emigration and depopulation [11].

The interest that the territorialised agrifood systems and the landscapes have generated, together with the multiple epistemological and regulatory meanings and the diversity of contexts and work situations, invites reflection on their meanings and importance.

This article has three objectives: (i) to explore the epistemological links between territorialised agrifood systems and landscapes; (ii) to examine in detail the regulations and instruments capable of articulating proposals for territorial development that combine the logic of both landscapes and territorialised agrifood; (iii) to determine whether the United Nations Food and Agriculture Organization's (FAO) Globally Important Agriculture Heritage System (GIAHS) is capable of articulating these proposals in rural areas with depressed economies.

The research starts from the following assumptions regarding territorialised agrifood systems and landscapes:

- The indispensable consideration of the agricultural component being anchored in territories, which cannot be ignored from the optic of landscapes and is not always considered by studies on territorialised agrifood. This being anchored in territories has an impact on landscapes' sustainability and is related to biodiversity and the biodiversity of agricultural systems.

- The necessary establishment of the links in the agrifood–production–processing–distribution chain in these systems' demarcation areas or surrounding areas; only consumption should have an exogenous or distant location.

- The demand for a management instrument with the capability to articulate the system's own socio-institutional networks [1,2] and to channel their expectations of development. This instrument would have to fulfil two requirements: (i) be organised from the local level up; (ii) combine the ways that the appearance- and preservation-based future planning for landscapes is presented with the perspectives for territorialised agrifood aimed at socio-economic functions and ends from the perspective of sustainability. The FAO GIAHS programme is considered to be the instrument that could do this in depressed rural areas.

In empirical terms, the analysis refers to the Axarquía supra-municipal district (Malaga province, southern Spain), which is chosen as an example to test the premises of the research. This is a mountainous area in the hinterland of one of the main international tourism enclaves (the Costa del Sol) and situated alongside the Malaga urban agglomeration, the fifth largest in Spain in demographic terms. Despite being in decline, the vineyard and the dried grape, or raisin, productive systems pervade the character of the district's landscapes and offer prospects for development. The raisin has been protected by the Pasas de Málaga Protected Designation of Origin (PDO) since 2013 and the FAO recognised the GIAHS Malaga Raisin Production site in 2018.

## 2. Materials and Methods

This article is structured into two phases, theory and a practical exemplification (Figure 1):

Phase 1. The first phase is theoretical and twofold: epistemological and regulatory (Section 3.1). It details the conceptual and prospective frameworks of landscapes (Section 3.1.1) and territorialised agrifood systems (Section 3.1.2) and focuses on the GIAHS programme as the nexus between the two.

The following R&D&I projects have been used to recap the concept of landscape: Spanish Agricultural Landscapes (SEJ2006-15331-C02-01, 2005–2009 and CSO2009-12225-C05-05, 2010–2014), Heritage Landscapes (CS O2012-39564-C07-07, 2013–2015) and listed UNESCO World Heritage Cultural Landscapes (CSO2015-65787-C6-6-P, 2016-208), and their findings [12,13].

The concept of territorialised agrifood systems was re-examined in the framework of the R&D&I Spanish multifunctional and territorialised agrifood systems project (SAMUTER) (PID2019-105711RB-C62, 2020–2022). The scientific literature available on Google Scholar and Scopus was checked and the following keywords were used as search arguments: 'territorialised agrifood systems' and 'localised agrifood systems'.

**R&D&I Projects**

*Spanish Agricultural Landscapes*

*Heritage Landscapes*

*Cultural landscapes on the UNESCO World Heritage List*

*Multifunctional Territorialised Agrifood Systems in Spain*

Institutional Frameworks
-International (FAO; Unesco; EU)
-National/regional (ministries and regional departments with powers)
-Local (provincial governments, municipal consortia, and town halls)

**Phase 1. EPISTEMOLOGY AND REGULATIONS**

**LANDSCAPE**

**AGRICULTURE AND TERRITORIALISED AGRIFOOD**

**GIAHS**

Sources

Cartographic, documentary, bibliographic and statistical

Fieldwork

Territorial recognition

Semi-structured interiews

**Phase 2. EMPIRICAL: The Axarquía vineyards (Malaga province, Spain)**

**SELECTION CRITERIA**
**-Recognized by GIAHS**

**-Protected Designation of Origin**

**-Exceptional landscapes**

**-Systems of high nature value**

**-Mountain area**

**Figure 1.** Methodology and research sources. Source: prepared by authors.

The programmes and regulations were examined at different levels: international (FAO, UNESCO, and European Union), state and regional (ministries and regional departments responsible for agriculture, food, spatial planning, the environment, and heritage), and local (provincial governments, municipal consortia, and town halls). The websites of these institutions were visited and the Google search engine was used with the keywords 'landscape', 'territorialised agrifood systems', and 'GIAHS'.

Epistemological reviews enabled to observe the points of convergence and discrepancy between landscapes and territorialised agrifood and also identified the most common empirical and methodological approaches. The concept of character was key in landscape [14]. In their case of territorialised agrifood systems, detailing their attributes (proximity, governance, and sustainability) was a determinant.

The examination of regulations and programmes recommended a focus on the study of GIAHS as a legal entity that encourages the dynamic conservation of territorialised agrifood systems and contributes to the preservation of landscapes and the development of sustainable agricultural practices. The FAO website was essential in all these cases [15].

Phase 2. The second phase empirically exemplifies the theoretical premises and focused on the Axarquía vineyards in the province of Malaga (Andalusia, Spain) (Section 3.2.). This case was selected based on several criteria: (a) The vineyards are recognised by GIAHS; (b) they possess a certificate of Protected Designation of Origin; (c) their landscapes are exceptional; (d) their land uses are typified of high nature value; and (e) they are located in a mountainous area.

With respect to the structure of this empirical part of the study, after a brief presentation of the area (Section 3.2.1.), its landscapes are examined in detail (Section 3.2.2.), and the agrifood system of the raisin is described (Section 3.2.3.); both aspects are related to the landscapes' and agricultural systems' conditions of sustainability and agrobiodiversity attributes. Lastly, an analysis is conducted of the GIAHS programme and the forecasts for territorial development associated with the programme (Section 3.2.4).

Multiple procedures and sources were used in the empirical part of the study:

(a) The 'La Axarquía Malaga Raisin Production System' application file was a key document [16];

(b) Cartographic sources and fieldwork were used to analyse the specific landscape structure. The former included the Andalusian Landscape Map [17]; the Malaga Provincial Catalogue of Landscapes [18]; the Spanish Atlas of Agricultural Landscapes [12]. Territorial recognition consisted of a five-day visit to the area (4 to 8 December 2021);

(c) Documentary, bibliographical, and statistical sources that expressed the main attributes required of these systems (proximity, sustainability, and governance) were used for the analysis of the agrifood system. Indicators of the localisation of the agricultural production and agro-industrial phases were used to rate proximity. Sustainability was evaluated with indirect (agrarian systems of high nature value) and direct indicators (surface data and organic operators registered with the Andalusian Organic Production Reporting System (SIPEA)) [19]. Both of the indicators are linked to the sustainability conditions and the quality of biodiversity and agrobiodiversity.

(d) The governance system and its programmatic and territorial development forward planning merited special treatment. For their analysis, qualitative research techniques were used (semi-structured interviews). Ten interviews were given to stakeholders between 25 November 2021 and 9 February 2022. The interviewees were selected from among the groups of actors involved in the GIAHS to enable the identification of the specific positions of each: farmers (2 farmers from Almáchar and the Union of Smallholders of Malaga); the agrifood industry (San Isidro Labrador Farmers Cooperative and the UCOPAXA Farmers Cooperative); regulatory organisation (Malaga Raisin PDO); representatives of the public administrations (Axarquía Muncipal Consortium; Andalusian Agriculture and Fisheries Agency); development agencies (the Association for the Promotion of Tourism in Axarquía); and Civil Society (The Muscat Grape Association). Before examining the interviews, we formulated codes or primary categories based on the questionnaire question guide; the theoretical framework was also used as a reference for the creation and search for concepts. The codes used to treat the content extracted from the interviews were: 'identification and evaluation of landscape components'; 'roles of the agrifood system'; 'links between the production and processing stages'; 'connections between the landscape and the agrifood system'; and 'evaluation of the territorial development model proposed by GIAHS'. Apart from the attributes of territorialised agrifood systems, the obtained responses were also related to sustainability, governance, and proximity. Figure 2 gives the list of agents interviewed and the bases of the questionnaire.

| Agents interviewed | Questionnaire | |
|---|---|---|
| - Interview 1: Farmer 1 from Almáchar<br><br>-Interview 2: The Union of Smallholders of Malaga<br><br>-Interview 3: San Isidro Labrador Farmers Cooperative<br><br>- Interview 4: UCOPAXA Farmers Cooperative<br><br>-Interview 5: The Muscat Grape Association<br><br>-Interview 6: Association for the Promotion of Tourism in Axarquía<br><br>- Interview 7: Malaga Raisin PDO<br><br>-Interview 8: Axarquía Municipal Consortium<br><br>-Interview 9: Andalusian Agriculture and Fisheries Agency<br><br>-Interview 10: Farmer 2 from Almáchar | **Landscapes**<br>-Identification and evaluation of components and unique features<br><br>**Agrifood system**<br>-Economic, production and non-production roles<br>-Contribution of agriculture to sustainability<br>-Conditions of agricultural biodiversity<br>-Connections between production and processing links<br>-Brief diagnosis of system<br><br>**GIAHS**<br>-Viability and pertinence of GIAHS for articulating development in the area from the perspective of sustainability<br>-Evaluation of Action Plan and its contribution to sustainability and biodiversity; of the actors and their involvement<br>-Evaluation of the project's current situation to articulate territorial development based on the concept of sustainability | **Trasversalities**<br><br>-Connections between landscape and the raisin agrifood system from the perspectives of sustainability and agrobiodiversity<br><br>-How the Action Plan contemplates landscapes<br><br>-Evaluation of the integrated, sustainable territorial development model and its mechanisms to articulate between landscapes and the agrifood system |

**Figure 2.** Interview methodology: agents and questionnaire. Source: Prepared by authors.

## 3. Results

### 3.1. Landscapes and Territorialised Agrifood Systems. Epistemological and Regulatory Approaches

3.1.1. The Ways Landscape Is Understood

The landscape is a polysemic, complex, and dynamic concept related to the appearance of land configurations and their aesthetic projections. Farming, an activity rooted in the day-to-day and hard toil, is not consonant with the initially elitist and contemplative vision of landscapes. The European Landscape Convention (hereafter, the Convention) [14] marked a change—at least on paper—in its reference to all landscapes: 'It concerns landscapes that might be considered outstanding as well as everyday or degraded landscapes' (art. 2) and its definition of a landscape as 'an area, as perceived by people, whose character is the result of the action and interaction of natural and/or human factors' (art. 1).

Every landscape's character, or hallmark, is a key notion. Landscape Character Assessment (LCA) [20–22] takes the methodology that it develops from landscape taxonomies inspired by atlases and catalogues [12,13,17,18]. In its references to specific areas, the character equates to something akin to each landscape's fingerprint, which comes from every area's unique and unrepeatable combination of: (i) a natural medium (topographic, geological, climatic, edaphic, etc.); (ii) some processes of historical construction (expressed in settlements, transport links, land use, plot structure, etc.); and (iii) a perceptive framework (social and institutional) that identifies, appraises, and manages landscapes and invests them with the qualities of heritage and identity. This turns the landscape and agricultural

landscapes into a symbiosis of nature and culture in all senses of the word (social links, traditions, imaginary, etc.) [23].

The Convention also urges the protection, planning, and management of landscapes (art. 1). This represented another change from their initial assimilation to exalted and picturesque spaces and objects of conservation. From this derives the inclusion of agriculture in territorial landscape management, but agrarian policies rarely form part of landscape management instruments. Agricultural uses are perceived as an extension, not as their production dimension. Spatial planning is used to mitigate landscape deterioration and a lack of organisation, especially in periurban areas [24]; the preservation of agriculture with a particular cultural and/or natural value thanks to its recognition as international heritage (UNESCO World Heritage Convention cultural landscapes; MaB Programme Biosphere Reserves, etc.) or by the state (assets of cultural interest and protected natural spaces) [25]. Agricultural production management, inescapable from the perspective of territorialised agrifood systems, rarely forms a part of landscape studies and interventions. This results in a kind of territory where agriculture (understood as an extension) has no links to its functional and production bedrock.

### 3.1.2. Approaches from the Territorialised Agrifood System Perspective

The notion of the territorialised agrifood system is broad-based. It has multiple epistemological bases [26] that share a sustainability, vindicatory, alternative, and redemptive component [27–29]. Internally, these are systems composed of agricultural, agro-industrial, and agro-commercial, etc., threads that are conventionally treated individually but which are now perceived through the prism of sustainability and the consideration of food as a production-processing-distribution-consumption continuum [30]. Another basic change is the descriptor of 'territorial'/'territorialised' that is associated with them, with different meanings and a common denominator: the assimilation of said 'territoriality' into alternative patterns to contend with the outrages of intensive farming and the imbalances of neoliberal globalisation and as a kind of antidote to overcome these [31].

The status of territorialised system entails the geographical tethering of the agricultural production and agro-industrial phases, as well as of other links in the food chain [32–34], which again refer back to its sustainability. From this arise some of its attributes: (i) proximity, expressed in agro-ecological practices [6,35]; in sustainable short circuits (of not more than one intermediary between producer and consumer) [36–38]; in cultural foundations underpinning foods that identify places and that involve consumers with producers [37]; and (ii) governance, understood as a social and institutional arrangement in locally driven projects [33,39,40].

Some other implicit defining attributes are: (iii) sustainability, associated with environmental externalities [6,41], agrarian systems of high nature value [7,8], protected natural spaces [42], and landscapes [43,44]; and (iv) multifunctionality, determined by the roles that agriculture plays in agrifood systems and in the development of the territories that embrace them [3–5].

Apart from epistemological endorsement, territorialised agrifood systems have a future-oriented (and regulatory) focus [7]. The institutions, regulations, and programmes involved in their management have expanded: from agricultural policies to other food-chain regulatory instruments (e.g., Spanish Law 12/2013, concerning the improvement of the Food Chain) [45] and to the point of concerning institutions and programmes with no previous connection to this area (spatial planning, preservation of natural and cultural heritage, etc.).

Agricultural policies continue to play the determinant role, especially the CAP (Common Agricultural Policy), which is imbued with environmental importance that derives in territorial and landscape nuances. Its assumption, via the ecoschemes, of the European Green Deal and its Farm to Fork and Biodiversity Strategies [9,46,47] and its transfer to state strategic plans (in Spain, the National Strategic Plan for the CAP post-2020) [48] express the above. The outcome is a varied scenario of territorialised agriculture that guarantees

(with many nuances) the maintaining of biodiversity but which only addresses landscapes partially, without taking into consideration all of the landscape proposals advocated in the Convention.

### 3.1.3. GIAHSs as an Agrifood and Landscape Articulation Instrument

The GIAHS programme emerged in 2002 at the 39th World Summit for Sustainable Development [15,49,50]. Since then, many analyses have been conducted on this concept, especially in Asia and the Pacific [51–53]. Studies have been less numerous in Europe, the Near East, Africa, Latin America, and the Caribbean, where there are few GIAHS sites [50,54].

GIAHSs are aimed at agrarian systems of high natural value managed by de-capitalised family smallholdings and, by extension, at the depressed rural areas, where these types of farms are found. For the main part, these are smallholdings and areas with agricultural limitations that lack the structure and finances to implement modernising processes. These circumstances have allowed them to retain some resources that have today gained in value: production capital with its roots in the land (traditional farming that supplies quality food); gene banks (autochtonous breeds and varieties); and natural (agrobiodiversity) and cultural heritage (centuries-old agricultural practices, accumulated knowledge, exceptional landscapes, etc.). Many hurdles have to be overcome for these to be brought into action: abandonment and depopulation, lack of initiative, little social capital, inappropriate policies and few incentives, deficient infrastructure, territorial pressure, etc.

The purpose of the GIAHS programme is, precisely, to contribute to activating these resources and mitigating obstacles in various ways: (i) raising local communities' awareness of the values of traditional types of agriculture; (ii) in agreement with said local communities, articulating development projects that integrate all their resources, both production (agricultural, livestock-farming, and agro-industrial development) and non-production (enhancing the value of their landscapes through rural tourism and agrotourism); and (iii) giving technical advice.

Figure 3 synthesises the demands required of GIAHS candidates. The links between territorialised agrifood systems and landscapes from the perspective of sustainability are evident. 'Dynamic conservation', which alludes to reconciling conservation and development, is a key concept. Locally managed governance is another basic principle. This is expressed in the 'Action Plan' or 'Dynamic Conservation Project' that applicants must provide, which must contain an express indication of the resources that are intended to be operationalised, the institutions and agents involved, the policies and instruments available, and the anticipated outcomes.

### 3.2. Landscapes and Territorialised (Agri)Food in Axarquía

#### 3.2.1. The Study Area

The Axarquía supra-municipal district is located in the eastern sector of Malaga province (Spain) (Figure 4). It covers an extensive surface area (102,800 ha) on ascending land that stretches from south to north between the Mediterranean Sea and the Alhama, Tejeda, and Almijara mountains (circa 2000 metres in height).

The district's altimetric gradation entails major changes in land occupation and uses (Figure 5). Three imaginary zones with growing homogenizing trends can be differentiated:

- To the south, a very periurbanised area of tourism expansion, with the presence of some quite large urban nuclei (between 20,000 and 80,000 inhabitants and Vélez-Málaga as the administrative capital); intensive agriculture (subtropical crops, mostly avocados and cherimoyas on terraced slopes) is the main agricultural use [55].
- A central strip of steep-sided mountain foothills. There are numerous population nuclei of reduced size (between 300 and 2000 inhabitants and with a slowing regressive demographic dynamic) [56,57]. This intermediate strip is the area where the vineyards are located in a crop mosaic with other uses (olive trees, cereal crops, almond trees,

forestry exploitations, etc.) that is today increasingly being encroached upon by rising periurbanisation and subtropical crops [58,59].

- In the southern sector (Periana, Álora, Canillas de Aceituno), the slopes are gentler, and olive groves are more prevalent.

The study area, which corresponds to a GIAHS area (28,039 ha, 20 municipalities and 124,601 inhabitants) is characterised by slow demographic growth (2020–2010: 5%) that is negative in some municipalities and an ageing population (20% of the population over 65 years of age) [60].

### 3.2.2. The Raisin as a Landscape and a Mark of Identity

In Axarquía, raisin grapes occupy a small surface area (1113 ha, rising to 2650 if the vines for Malaga wines are included) [16] and have been in decline since the crisis unfolded in traditional agriculture, with a 26% fall in the last 5 years alone [61]. Despite this, these landscapes are suffused with raisin grape farming, which marks their identity and plays a prime environmental role.

| Indication of relevance (historical and current) and of risks | |
|---|---|
| **Historical relevance** | Ability to adapt to the surroundings: through handed-down knowledge and specific techniques in current systems that shape their landscapes |
| **Current relevance** | Present and future ability to provide subsistence means that guarantee food security and quality of life to local communities: thanks to the mobilisation of all the resources in the GIAHS area (production, environmental and landscape) |
| **Risks** | Diagnosis of threats and challenges |

| Defining criteria of a GIAHS site's values |
|---|
| 1. Agricultural systems that contribute to food security and a means of subsistence |
| 2. Systems with high unique agro-biodiversity |
| 3. Traditional knowledge and techniques transmitted down the generations |
| 4. Strong cultural values, collective forms of social organisation and value systems for resource management and knowledge transfer |
| 5. Outstanding terrestrial and marine landscapes that arise from ingenious land and water management systems and technologies |

| Presentation of an action plan for dynamic conservation |
|---|

**Figure 3.** Requirements and criteria for GIAHS candidates. Source: Prepared by authors.

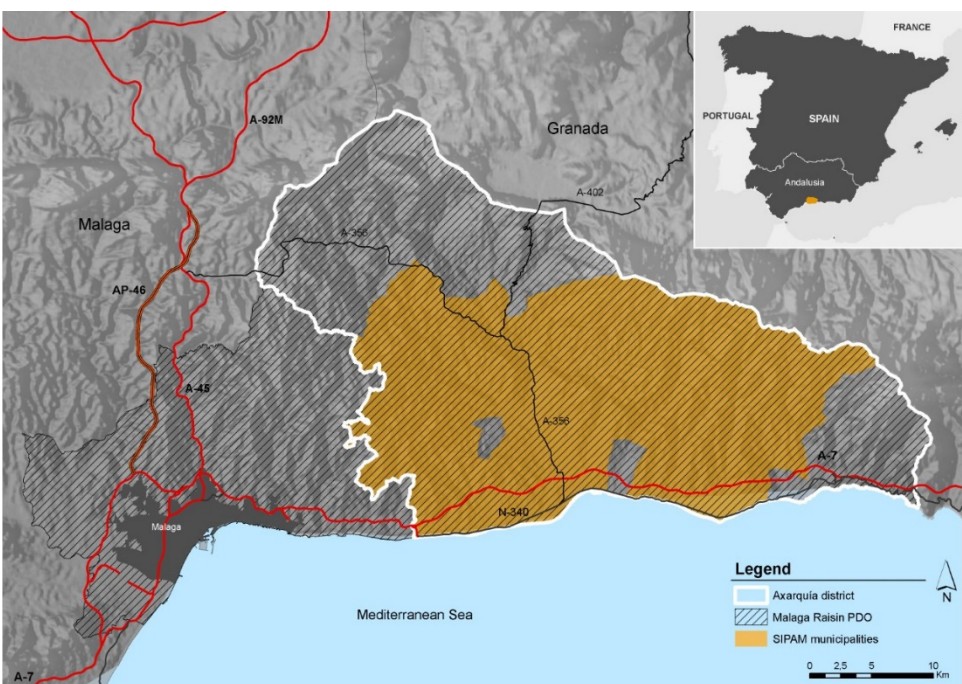

**Figure 4.** Location of Axarquía supramunicipal district. Source: Prepared by authors.

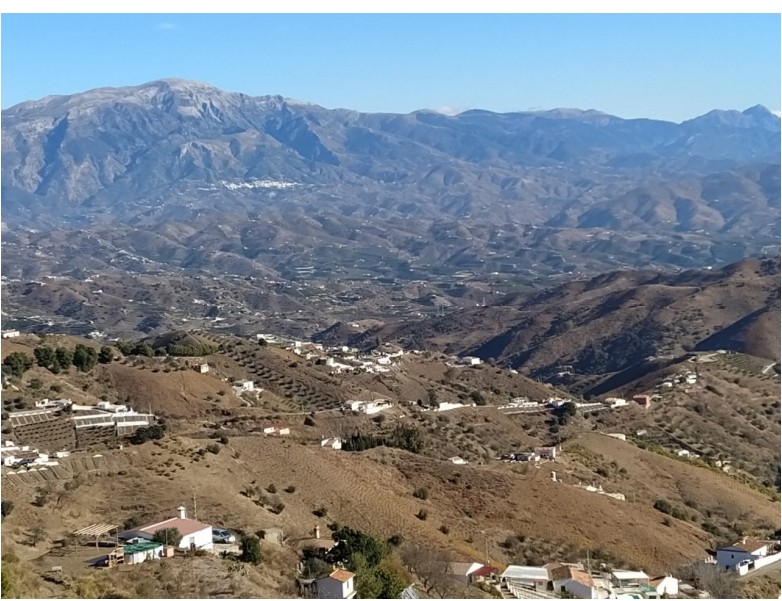

**Figure 5.** Orography and settlement in Axarquía supramunicipal district. Source: Authors.

The vineyard landscape is spread over a limited natural framework: mountains with steep slopes, a shallow soil covering (over shale and slate) poor and highly eroded nutrients, and a Mediterranean climate with scant yet torrential rainfall. However, it is well adapted to this environment and, along with olive and almond trees, is one of the few crops that grow here; in addition, it contributes to their improvement thanks to the retention of the edaphic layer, preventing desertification, acting as a haven for numerous species, etc. The resulting landscapes are spectacular: The uneven terrain improves visibility and the richness of the scenery (Figures 5 and 6); the phenological cycle of the vine (with leaves that come out in summertime) adds a touch of colour and a feeling of freshness to the torrid brown summer months [59,62].

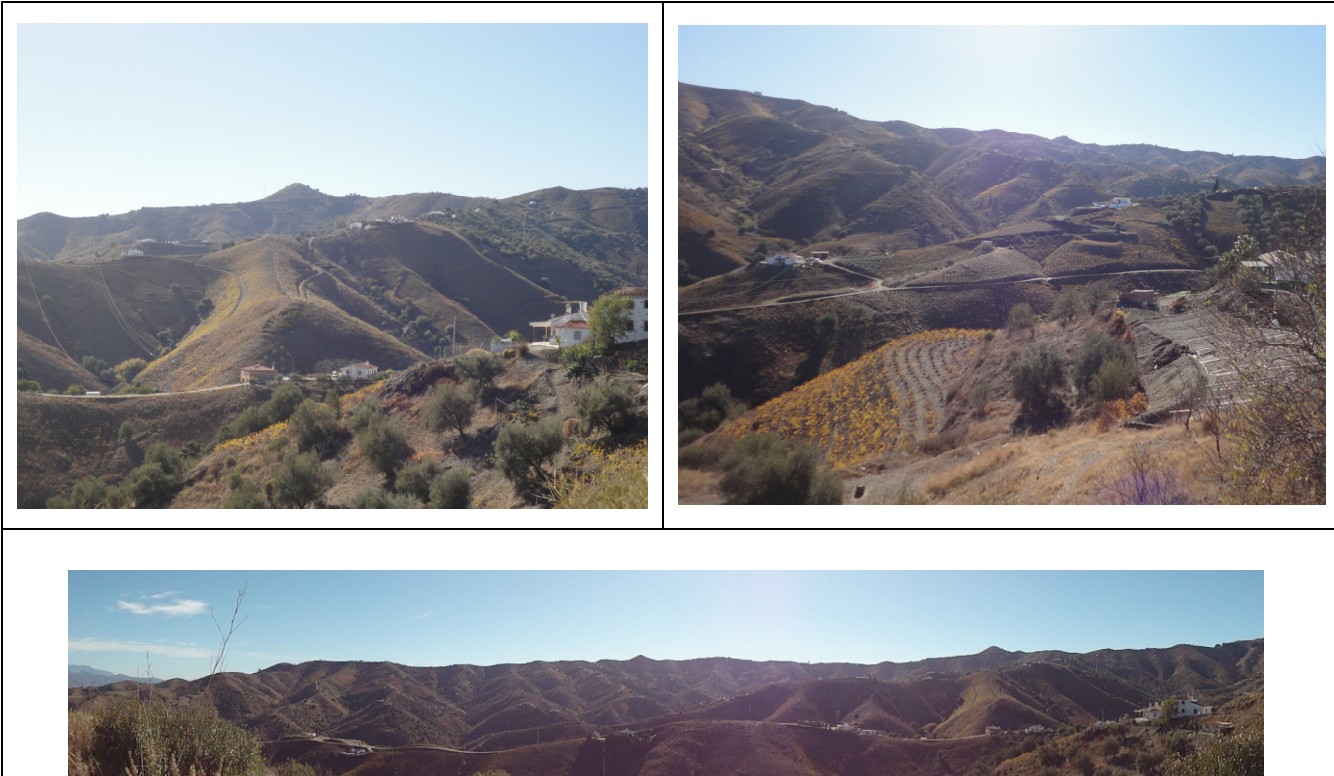

**Figure 6.** Vine-covered landscape in La Axarquía. Source: Authors.

History has yielded an exceptional agricultural landscape: suntraps covered in leafy green vines with unique local architecture of traditional *paseros* and *lagares* scattered around the vineyards (Figure 7). As a whole, this is a landscape of highly diversified mid-height Mediterranean mountains: a mosaic of vineyards and orchards dotted among ploughed fields, pastures, and natural brush; Andalusian 'white villages'; and an interstitial habitat of modest farm estates (with the characteristic *paseros* and *lagares*).

Interview(I) 1 'Our landscape is unique, and it's not only us who say that, it's been recognised internationally'.

I3 'Agricultural heritage is our hallmark because we look after the landscape'.

I7 'People are very attached to their birthplace . . . the landscape is linked to a population, an activity a product that breathes life into everything it creates around it'.

The raisin grape permeates the cultural roots and identity of Axarquía; it is the backbone of the region's social relations and generates a way of life that is a mark of the district's identity. Farming is associated with rites and festivals that hark back to life in the *lagares*. The cultural legacy is immeasurable: place names, refrains, songs and dances (*Las Candelas*, la *Zambomba*, *Los Verdiales*, *La Rueda*), and food festivals (the El Borge Day of the Raisin, the La Viñuela Raisin Festival, the Moclinejo Winemakers' Festival, etc.). The people involved are proud of their landscapes.

I1 'Families stay out of the love they have for the land and the product . . . it's all to do with culture and not many of the age-old themes have been lost, such as the "Sounding of the Conch Shell", "Dancing in the Round", the "verdial"- the typical song of Comares. This is all tied up with a product and a place'.

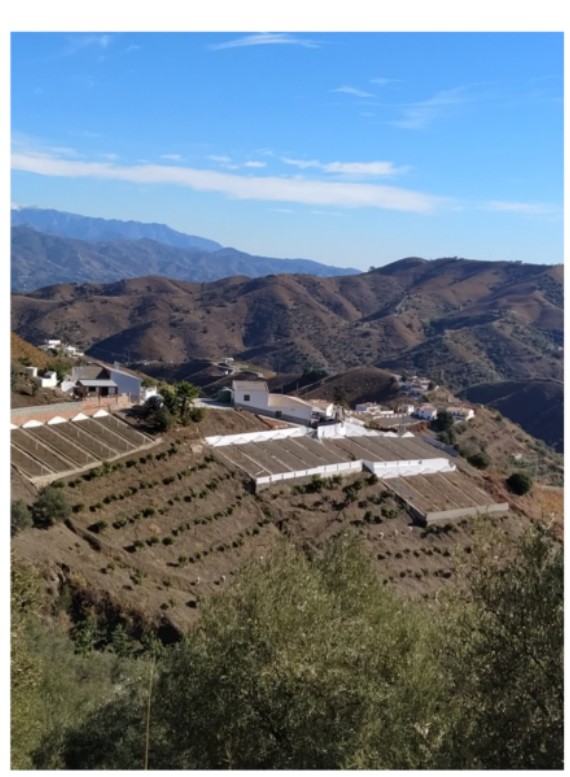
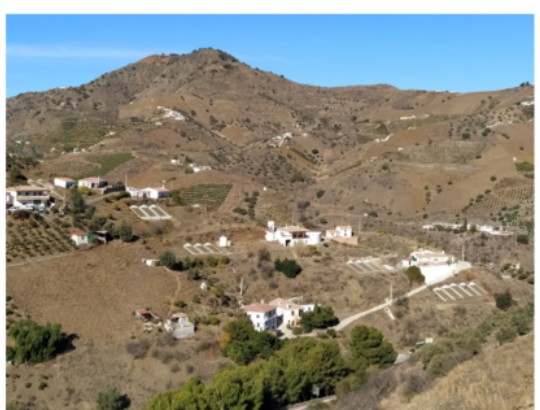
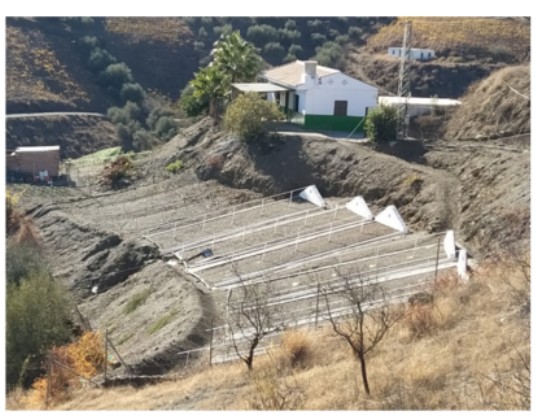

**Figure 7.** *Lagares* and *paseros*. Source: Authors.

### 3.2.3. The Raisin Grape Agrifood System

Axarquía is the main raisin grape growing area in Spain. The variety used is the Muscat of Alexandria grape, which is highlighted in the interviews as unique in the world for having three end uses (to make raisins, to make wine, and for use as a table grape).

I1 'The product's strong point is its quality. What the Muscat of Alexandria grape has got is that you can use it for three things: wine, raisins and to eat straight. No other variety can do that. That's its really big strong point: 'quality and variety''.

The structure of the raisin agrifood system is complex (Figure 8), and its fundamental features are fragmentation, governance, and proximity.

The agricultural production link is represented by family smallholdings (1.22 ha/farm and 0.41 ha/plot) [61].

I9 'With this terrain, the production system itself and the passing of history (plots divided through inheritance), the average plot size is very small, and that restricts applying an economy of scale'

Raisins are sun-dried on the *paseros* set out on the farm (5 *paseros*/farm for a total of 4593 *paseros*) [16]. The raisins are 'picked' from the stalks in the *lagares*, which are also on the farm itself. This guarantees the proximity of the agricultural production practices.

The wine grapes are processed in wineries (a total of 18 companies) [63] located in the towns in the area. The raisins are sorted by first-tier cooperatives and are packed and marketed by industries and a second-tier cooperative, UCOPAXA. This is made up of 8 first0-tier cooperatives and 700 farmers and is the backbone that holds together this system, predominantly composed of smallholdings. All the establishments are registered in Axarquía, which guarantees the proximity of the agricultural and agro-industrial links.

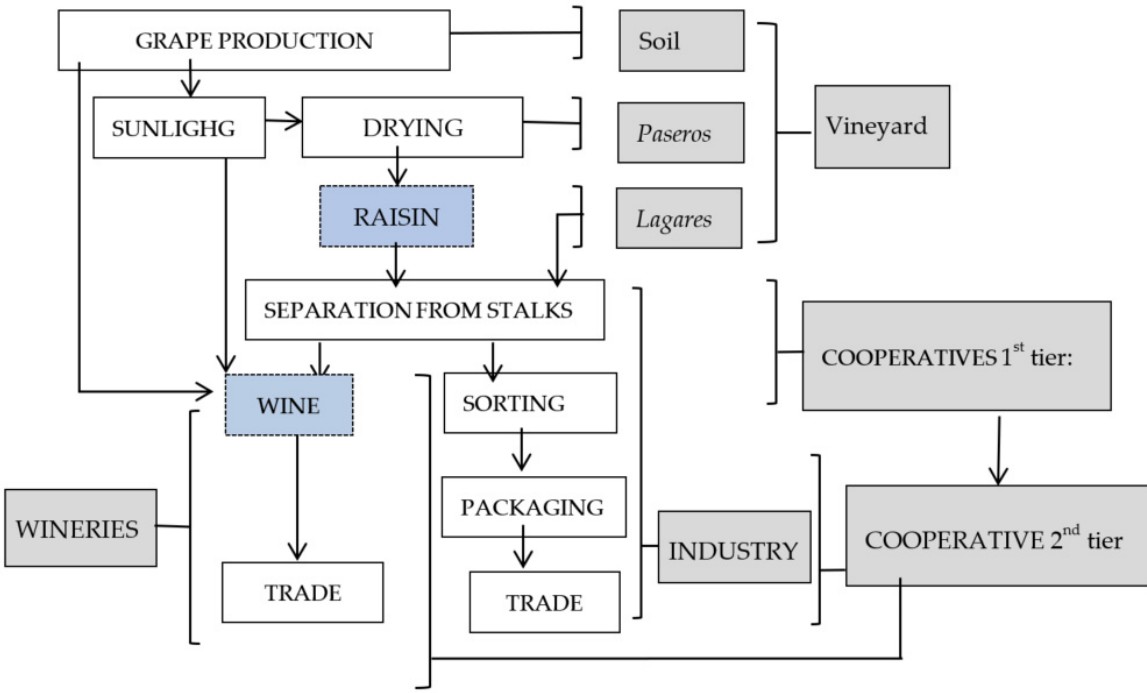

**Figure 8.** Raisin agrifood system. Source: Prepared by authors.

The Malaga Raisin PDO plays a key role in the entire system. It is used to commercialise 85% of Axarquía's raisin surface area (a total of 942 ha) [61]. Along with the Malaga and Sierras de Malaga PDOs, which are specifically for wine, the Malaga Raisin PDO is another indicator of territorial anchoring in the agricultural and agro-industrial components. It should not be forgotten that PDOs certify quality produce that fundamentally demand that all the agricultural and agro-industrial phases should be located in the demarcation area; their mere presence can be taken as an indicator of the proximity attributes required of these systems.

Sustainability is expressed in the landscape: The sloping land enables the air to circulate around the plant without the need to apply any phytosanitary protection; the vines hold the soil together and prevent erosion; the soil underlain by slate offers up some wines with some very special organoleptic characteristics.

The farmwork is manual, and the type of terrain means that any hauling is performed using animals. Apart from benefiting sustainability, this contributes to the production system interlocking with its landscapes. The interconnection of the agrifood system and the landscape is again seen to be unbreakable.

Another exponent of sustainability is grape farming itself: The specialist literature states that grapevines are one of the agricultural systems of high natural value [8]. The insignificance of organic farming according to official data is surprising: 3.16 ha of table grapes and 4.93 ha of vines for wine [19]. The third-party certification processes are bearable for an intensively worked family farm system with low economic performance and low capitalisation.

### 3.2.4. Action Plan for the Dynamic Conservation of the La Axarquía Malaga Raisin Production System

The Action Plan is a key document in terms of prospects, sustainability, governance, and territorial development. It contains five lines broken down into measures (Figure 9) that materialise in the form of actions and projects.

| Strategic lines | Measures |
|---|---|
| Line 1 concerning maintaining and improving the production system | Upkeep of *paseros*; continued use of means of production; technological improvements; actions to prevent the abandonment of plantations |
| Line 2 concerning improving commercialisation | Promotion; company synergies and collaborations; product diversification; new forms of commercialisation |
| Line 3 concerning conserving and handing down the agricultural heritage | Craft linked to the production system; preservation and passing down of production heritage and cultural wealth |
| Line 4 concerning tourism promotion | Tourist itineraries; gastronomy; integration of citizens into agricultural work |
| Line 5 concerning research and dissemination | Research; preparation of reports; recognition as a GIAHS |

**Figure 9.** Strategic lines and measures of the Malaga Raisin GIAHS Action Plan. Source: Prepared by authors, based on [16].

The implementation period for the plan is five years. The first meeting of the Axarquía GIAHS management body took place at the end of November 2021 (several of the agents interviewed insisted on a postponement). It is still too early to evaluate the level of execution, but the analysis of the plan itself, and the results shown in the interviews, allow for a glimpse of some aspects:

- The most advanced actions are research and dissemination; some have even concluded, including the inventory and mapping of the *paseros* and the drafting of the application file, as actions proposed in Line 5 [16].
- Promotion and tourism promotion activities and mobilisation are at an advanced stage (Lines 3 and 4): joint promotion of events (Muscatel Market); attendance at national and international fairs; the relaunch of raisin routes and itineraries and the projection of viewpoints; the strengthening of cultural facilities (La Axarquía Raisin Study Centre; Raisin Interpretation Centre and Raisin Museum); studies of tourist markets and relaunch of the La Axarquía brand; restoration of *lagares* for rural accommodation. This advance is to a great extent due to these actions having been initiated before GIAHS recognition.

I7 'The landscape is paramount [ . . . ] The tourism you have to attract to this area is the kind of tourism that is looking for experiences, that wants to go somewhere that they can understand and experience. And landscape is an experience, you can live it'.

I8 'It's also important to highlight its importance for tourism because of the traditional means of production and its appeal to a variety of profiles. It can be an activity that absolutely respects the landscape and raises joint awareness of the product and the setting. There's a growing number of initiatives of this type that allow the enjoyment of healthy, natural spaces with a range of possibilities, including having these types of experiences and learning to respect the environment'.

The interviewees consider that GIAHS offers an opportunity to improve the system, above all as a complementary revenue stream through the valorisation of these landscapes. The connection between the agrifood system and the landscape is well accepted.

I2 'In our opinion, it's very positive, it creates a synergy and wealth at both the promotion level and for the appreciation of La Axarquía's products'.

I7 'FAO has recognised the landscape, the product, the culture and the way of life . . . everyone is for GIAHS'.

At a more initial stage of execution are the projects related to raisin grape production (replanting with new vines), promotion, and commercialisation (Lines 1 and 2).

The prototype of the awning for the *paseros* (an innovation that is considered essential) is still pending.

I1 'Landscape conservation is going to depend on whether more vines are planted or not; it's one of the goals of the Action Plan'.

I5 'The mechanical awning is really important. But how's it going to be paid for? These people are smallholders and they haven't got any money to invest in awnings'.

I2 'The Provincial Government has commited to financing the prototype, but what's got to be done now is to get it to the farmers at zero or very low cost'.

The use of the GIAHS logo to promote production is still pending; there is little information in this respect. The anagram is used to signpost routes but not to promote the raisin (together with the PDO seal). The latter requires Ministry of Agriculture approval of the use of the symbol of GIAHS site production that already exists in other countries.

I4 'It's being looked at [use of the GIAHS anagram]. Phrases can now be used on packaging that refer to GIAHS, but the FAO anagram can't be used for commercial use, either'.

Farmers have been involved in the GIAHS candidature and feel they are represented in the management body. They perceive GIAHS recognition as an instrument to perpetuate the crop but complain that the actions are more focused on promoting tourism than on supporting agriculture, and they find that disappointing.

I1 'We [farmers] have always been actively involved in the GIAHS. In the candidature and in the GIAHS Association'.

Despite the above, they complain that the actions focus more on promoting tourism than supporting farming; they find this disappointing.

I7 'GIAHS is a chance for the raisin to live on . . . The administration is slow, that's why farmers don't believe in it'.

I6 'farmers are front and centre, but they haven't profited in any way as yet'.

I1 'The farmers tell us it [GIAHS] hasn't got to them yet'.

I5 'As yet, we haven't benefited much from GIAHS'.

Strengthening governance has been one of the main achievements. Before GIAHS was declared, numerous agents were involved in the management and future plans for territorial development; this shows that the district already had relatively well-developed social capital.

Social capital has been strengthened by GIAHS recognition (Figure 10). The before and after comparison shows the following:

- Some actors have been strengthened (Municipal consortium and Provincial government).
- The importance of the Muscat Grape Association. This association represents civil society in the district and defends the continuity of raisin grape production and the conservation of its heritage. It was behind the GIAHS candidature.
- The Association for Tourism Promotion of La Axarquía also plays an important role.
- The incorporation of outside actors: University of Malaga, Costa del Sol Tourism and Planning Company, the Spanish Government's Department of Agriculture, and the Spanish Government's Local Office.
- Creation of a new actor, the GIAHS Association.

GIAHS has strengthened the links between the administrations, institutions, and stakeholders and between these and the outside. They have increased in number; networks are denser, and the system has been afforded a certain degree of internationalisation through the project for the Valorisation of Globally Important Agricultural Heritage Systems (VALSIPAM).

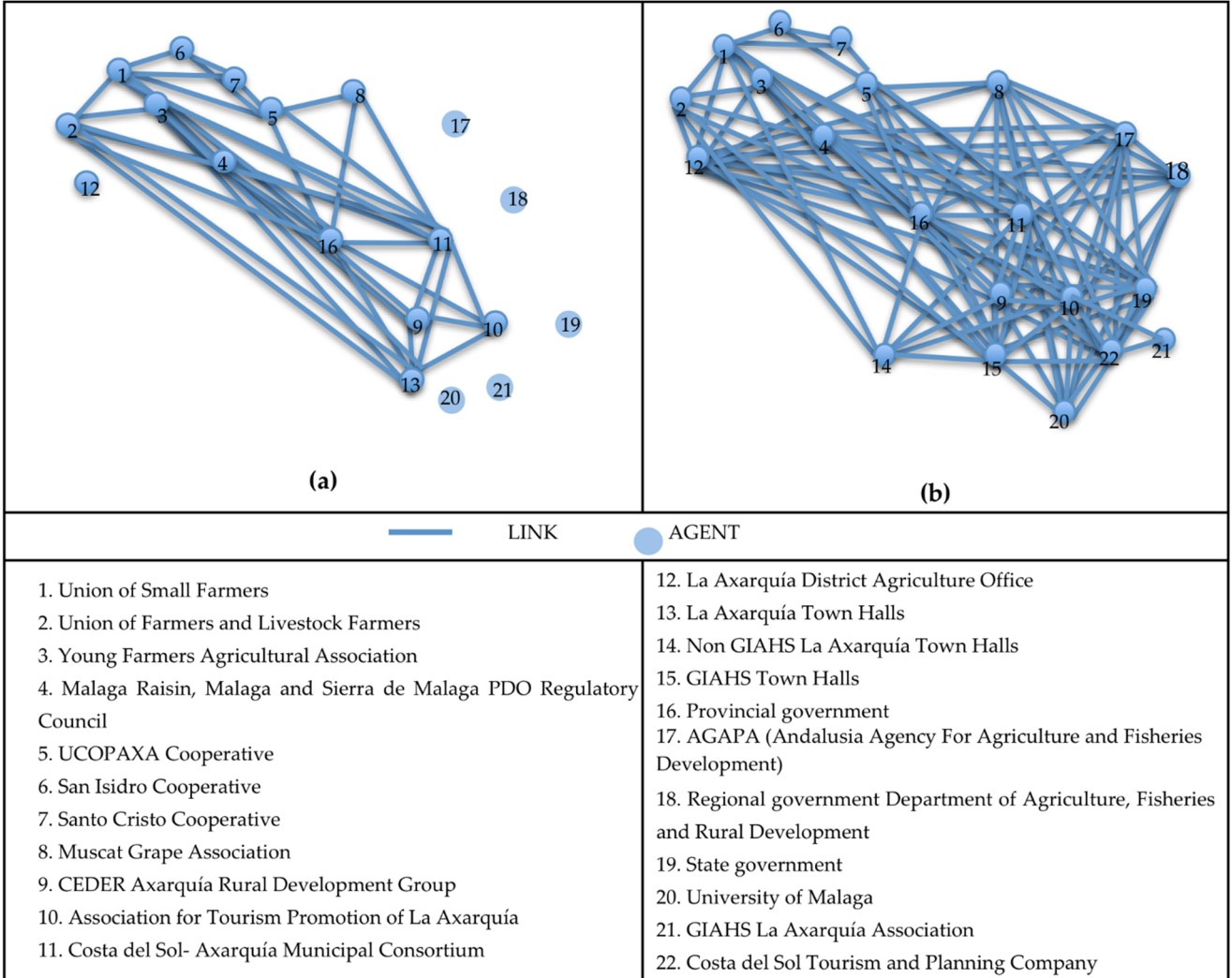

**Figure 10.** Social capital in La Axarquía (stakeholders, institutions and administrations, and links):
(**a**) Before GIAHS; (**b**) After GIAHS. Source: Prepared by authors.

### 4. Discussion

(a)  Landscape and territorialised agrifood systems. Points of discrepancy and convergence.

The landscape and the territorialised agrifood systems are preserved on different, albeit converging, epistemological and regulatory planes. The nexuses between them are very subtle and as yet unexplored.

The European Landscape Convention confers landscape status on agriculture. This represents a change from the previous assimilation of landscapes as exalted and picturesque areas; the change is less evident in regulatory terms. The future of landscapes is directed at landscape creation and planning [64] and the preservation of some landscapes (in special plans; in declarations of protected natural spaces and property of cultural interest) [25]. The management of agricultural production rarely forms part of landscape studies and interventions. The result of this is a kind of landscape with no farms and no farmers.

Studies on territorialised agrifood emphasise the anchoring of agricultural and agri-food production in territories, as well as sustainability, governance, consumption, and proximity trade [27–30]. Landscape, as a scenario that includes agriculture, is beginning to receive attention, particularly from the CAP, and is being assimilated to the environment and embodied in isolated components (small remnants of natural vegetation, terraces, plot boundaries, etc.) but rarely for itself. The concept of landscape championed by the European Landscape Convention has not been taken on board.

This article defends the unitary treatment of territorialised agrifood (in all its attributes: food sovereignty, local knowledge, cultural roots, governance, social capital, etc.) and landscapes and elaborates on the nexuses or relational arguments:

1. The first of these is agriculture itself, a word that has fallen into disuse, eclipsed by the notion of the agrifood system, which appears to seek to avoid the primordial meaning that alludes to the cultural value of the res agraria. This article flies the flag for agriculture: as a concept, as a fundamental component of territorialised agrifood systems, as an exponent of agrobiodiversity, and as the nexus with landscapes. Without agriculture (and farmers), there are no agrarian landscapes. This statement would appear to be obvious if it were not for the fact that an ever-growing number of studies focus on consumption and the last links in the food chain, in which the agricultural component is diluted or fades all together [65].

Not all types of agriculture (and the accompanying landscapes) possess the sustainability attributes of territorialised agrifood systems. Intensive industrial agriculture and multinational food companies would be excluded. So, the only types that would comply with this condition would be family farms; environmentally sustainable agriculture rooted in one's 'homeland' and its agro-organic bases and agriculture with landscapes that, despite their possible disorganisation and their loss of integrity and authenticity, arouse identity allegiances and locally managed agriculture.

2. The above refers to other close connections associated with the attributes of proximity and sustainability required of these systems [30]. Another key nexus is the presence of agricultural systems of high nature value, which are guarantors of biodiversity and projected in exceptional landscapes [7].

3. However, the attribute that best defines territorialised agrifood systems and that connects them to landscapes is governance. Territorialised agrifood systems are territories with a project, focused on agriculture and its landscapes, and articulated locally. Socio-institutional articulation is another major nexus; achieving this requires three circumstances [66]: the active involvement of the local public powers, their predisposition towards constructing shared scenarios for the future, and the existence of a legal management instrument capable of articulating the initiatives. In rural areas with depressed economies, with no relational capital or critical mass (due to emigration and depopulation), an external impetus is required. The epistemological and regulatory review leads us to believe that FAO's GIAHS programme could fulfil this need.

(b) The GIAHS programme as a territorial development instrument to articulate landscapes and territorialised agrifood.

This study confirms the virtuality of the GIAHS programme as an agriculture-based territorial development instrument. GIAHSs are shown to be an ideal instrument for the resilience of agrifood systems that, while unique with respect to their landscapes, traditions, crops, agricultural practices, and population, are beset by difficulties and threats such as ageing and the lack of generational succession, the land being abandoned, the lack of financial resources and urban pressure and other environmental sustainability issues [49,50].

I10 'Young people feel some attachment to the land, but they don't want to carry on with this. They just keep on with the raisins to top up their income'.

I7 'If the raisins brought in more profit, the young people would stay'.

Research pays special attention to the agricultural heritage, sustainable development, tourism, and biodiversity [67], as well as to the landscape [68], especially as a resource that can be mobilised for tourism purposes [69–71]. This results in the convergence of the logics of landscape and agrifood converge, which is a novelty with respect to the dominant ideas that recognise territories and landscapes with agriculture but without farmers. The attention that the GIAHS programme gives to tourism harks back to the above. This approach is, therefore, more complex and differs from the conservative notion of legal

protection such as UNESCO cultural landscapes [25]. GIAHS understands landscape in the sense that the European Landscape Convention does, but in its practical application, it goes beyond distinctive physical features and focuses on their functionalities (and economic dimensions); its preferential focus on agricultural systems of high nature value (vineyards, olive groves, rice paddies, traditional forms of irrigation, etc.) is an expression of the meaning and value of agriculture. The stress on selling agricultural landscapes through tourism detracts from the important role of agriculture.

(c)  Some considerations regarding the empirical analysis.

The supramunicipal district of La Axarquía is shown to be an excellent laboratory. The system woven around the raisin grape meets all the requirements of sustainability, proximity, and governance demanded by territorialised agrifood systems. The specialist literature includes vineyards among the agricultural systems of high nature value [7,8]; this implies the recognition of their role in maintaining biodiversity. The physical proximity of vineyards, *paseros* and wineries reinforces the conditions of sustainability and facilitates (physical and cultural) contact. Vines, raisins, and wineries permeate the character of these landscapes, establish the roots of their identity, and play a paramount environmental role.

Contrary to the results obtained in other studies, the Protected Designation of Origin has worked well as an indicator of proximity and territorial roots; this is what differentiates La Axarquía from other areas [72]. The same cannot be said of the organic surface area as an expression of sustainability. Its analysis using the data available in the SIPEA [19] has not delivered the expected results; this is a significant statement, given the role assigned to it as an indicator and argument to receive aid (such as from the CAP). This finding agrees with some studies that call into question the virtuality of organic agriculture with the proposal of other labels such as 'regenerative agriculture' [73].

GIAHS recognition has given a shake-up to the development of La Axarquía. The application file bears witness to the difficult situation that its agrifood system is going through and the urgency with which it must be addressed. Despite being targeted at agrarian systems of high nature value and farms, the emphasis that the programme puts on promoting tourism would explain the despondency of the farmers.

Governance, understood as concurrence and developed in the form of locally driven projects [32] has been confirmed in La Axarquía. In depressed rural spaces such as La Axarquía, with a depleted and elderly population and low incomes, it is absolutely essential that both the wide range of agents and the citizens should commit to their own development [30]. Different local and regional actors, the Muscat Grape Association and AGAPA (the Andalusian Agriculture and Fisheries Development Agency) especially, have played a fundamental role in promoting the district's application and its social capital has been fortified. The farmers have been identified as co-participants in the projects with representation on the management body; this is a difference from other analyses, which find discrepancies between theory and practice [74].

This was not an area neglected by the institutions or devoid of social capital. Despite the lack of coordination between them, there were numerous plans, projects, and recognitions prior to the declaration. Multiple examples confirm this: the Eastern Costa del Sol-La Axarquía Regional Spatial Plan [56]; the inclusion of Comares, Cútar, and Benamargosa in the registry of Andalusian Landscapes of Cultural Interest [75]; the inscription of several records of raisin grape growing in the Andalusian Atlas of Intangible Heritage [76]; the inclusion of La Axarquía and the Eastern Coast as Outstanding Landscapes in the Catalogue of Landscapes in the Province of Malaga [18]; the creation of the Costa del Sol-La Axarquía Municipal Consortium (1972); the forming of the Association for the Promotion of Tourism in La Axarquía (1987); the creation of the CEDER La Axarquía Rural Development Group (1992); the regulation of the Malaga Raisin PDO (1996); and the decree of a specific environmental measure for the 'Preservation of Unique Systems: chestnut and raisin' in the rural development programme schedule 2014–2020.

One of the GIAHS association's main achievements is the articulation of a proposal to integrate the initiatives around the raisin grape production system and raise awareness of a district body.

There are major achievements. Valorising GIAHS recognition is positive; the links and synergies between the raisin grape's production and landscape facets were repeatedly stated in the interviews. However, some concerns were also detected. Tourism is presented as a panacea; this coincides with what has occurred in other GIAHS areas [69–71] and in depressed rural areas in general. The interviewees perceive that tourism could be an efficacious tool to diversify the economy, as found in other studies [77] but that is no obstacle to doubting the sustainability of the tourism model [78] and its compatibility with the production model.

Considering that Axarquía has been studied as an example to test the research premises, that only some of the actions have been implemented, and the recent creation of the GIAHS management body, issues arise that have to be addressed in future research: (i) the compatibility of the actions with the criteria set out in the application; (ii) the contribution of GIAHS recognition to preserving and advancing the agrifood system; (iii) its effective role in the development and entrepreneurship in the district; (iv) whether the actions, once they start being implemented, really concur with the programme's flagship criteria of economic, social, and environmental sustainability; and (v) citizens' knowledge and perceptions of the GIAHS programme.

**Author Contributions:** Conceptualisation, R.S.-P.; methodology, R.S.-P. and G.G.-R.; validation, R.S.-P. and G.G.-R.; formal analysis, R.S.-P. and G.G.-R.; research, R.S.-P. and G.G.-R.; writing—original draft preparation, R.S.-P. and G.G.-R.; writing—review and editing, R.S.-P. and G.G.-R.; supervision, R.S.-P. and G.G.-R.; project administration, R.S.-P. and G.G.-R.; funding acquisition, R.S.-P. and G.G.-R. All authors have read and agreed to the published version of the manuscript.

**Funding:** Grant PID2019-105711RB-C62 funded by MCIN/AEI/ 10.13039/501100011033.

**Institutional Review Board Statement:** Not applicable.

**Informed Consent Statement:** Not applicable.

**Data Availability Statement:** The databases used in this research are referenced as sources: [19,60,63].

**Acknowledgments:** The authors would like to express their gratitude to all the people who have collaborated in this research.

**Conflicts of Interest:** The authors declare no conflict of interest. The funders had no role in the design of the study; in the collection, analysis or interpretation of data; in the writing of the manuscript, or in the decision to publish the results.

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
