# Peer review of "GIAHS as an Instrument to Articulate the Landscape and Territorialized Agrifood Systems—The Example of La Axarquía (Malaga Province, Spain)"

_land, doi:10.3390/land11020310_

Round 1
Reviewer 1 Report
The article is very interesting and seeks to show finally the links that exist between the landscape quality supported by an agriculture with a strong traditional component. This agriculture is supported by agrifood products with high added value, in particular raisins.
However, this work should also try to better prioritize the scale of values to better target the sustainability of these landscapes registered in GIAHS. In this respect, it is not surprising that farmers find the overemphasis on tourism disappointing. In terms of the theoretical basis that underpins the developments in this article, we can add the anthropological notion that we must avoid separating nature (agriculture) and local culture (social links and representations of the world). The priority from an anthropological point of view is the structuring of the agricultural community, not tourism.
That the GIAHS increases the links between institutions is certain, but this does not mean that it did not exist before, or at least there was certainly a community of interest.
There is a lack of biodiversity and agro-biodiversity assessments in areas where the landscape is still well maintained to support the claims of the article.
The survey methodology is not explained in detail. In particular, while it is clear that the stakeholders interviewed are involved, it is not clear how the data processing and transcriptions of the collected speeches are carried out. The sample of stakeholders only concerns the institutions a priori. The sample size could have been increased with ordinary citizens and people who are directly interested in the social connections they have with the agricultural community outside of tourism.
Figure 8 needs to be modified. It is cropped and the arrows are poorly drawn in some places.
Make difference between actors and stakeholders
After these modifications, I strongly encourage the publication of this article.
Author Response
"Please see the attachment"

Reviewer 2 Report
Dear authors
my suggestions/comments are as follows:
- the article is well structured and argumented
- the theoritical background is very good
- it would be good if there was a table with the percentages of crops in the total cultivated land in Axarquía
- when quoting interviews, it would be good to mention who says it. It is important who says what. For exemple it is important to know what the farmes say.
- line 414: "atomised": maybe the word is wrong
- line 464: "The first meeting ......". I think it would be better to change paragraph.
- line 607: "In urban areas .....". Please explain.
- line 582: "Without agriculture (and farmers) there are no agrarian landscapes". I strongly agree. In your rechearch, it seems that farmers are not involved in the project, according lines 519-526. I think you should emphasize it in the discussion. Without the active farmers' involment, the success of the projects is not guaranteed.
- I recommend presenting more results of your interviews and if possible to distinguish the views of different stakeholders.
Round 2
Reviewer 1 Report
The answers provided by the authors are appropriate.